# Reduced Carbon Cycle Resilience across the Palaeocene-Eocene Thermal Maximum

David I. Armstrong M[c]Kay*[1,a] & Timothy M. Lenton[2]

[1] Ocean and Earth Science, University of Southampton, National Oceanography Centre Southampton, Southampton, SO14 3ZY, UK (work undertaken here)

[2] Earth System Science group, College of Life and Environmental Sciences, University of Exeter, Exeter, EX4 4QE, UK

[a] Stockholm Resilience Centre, Stockholm University, Kräftriket 2B, SE-10691 Stockholm, Sweden (current address)

*Correspondence to*: David I. Armstrong McKay (david.armstrongmckay@su.se)

**Abstract.** Several past episodes of rapid carbon cycle and climate change are hypothesised to be the result of the Earth system reaching a tipping point beyond which an abrupt transition to a new state occurs. At the Palaeocene-Eocene Thermal Maximum (PETM) ~56 Ma, and at subsequent hyperthermal events, hypothesised tipping points involve the abrupt transfer of carbon from surface reservoirs to the atmosphere. Theory suggests that tipping points in complex dynamical systems should be preceded by critical slowing down of their dynamics, including increasing temporal autocorrelation and variability. However, reliably detecting these indicators in palaeorecords is challenging, with issues of data quality, false positives, and parameter selection potentially affecting reliability. Here we show that in a sufficiently long, high-resolution palaeorecord there is consistent evidence of destabilisation of the carbon cycle in the ~1.5 Myr prior to the PETM, elevated carbon cycle and climate instability following both the PETM and Eocene Thermal Maximum 2 (ETM2), and different drivers of carbon cycle dynamics preceding the PETM and ETM2 events. Our results indicate a loss of 'resilience' (weakened stabilising negative feedbacks and greater sensitivity to small shocks) in the carbon cycle before the PETM, and in the carbon-climate system following it. This pre-PETM carbon cycle destabilisation may reflect gradual forcing by the contemporaneous North Atlantic Volcanic Province eruptions, with volcanism-driven warming potentially weakening the organic carbon burial feedback. Our results are consistent with but cannot prove the existence of a tipping point for abrupt carbon release, e.g. from methane hydrate or terrestrial organic carbon reservoirs, whereas we find no support for a tipping point in deep ocean temperature.

## 1.     Background

The Palaeocene-Eocene Thermal Maximum (PETM) at ~56 Ma is considered a potential example of passing a tipping point in the carbon-climate system where a smooth change in forcing triggered a large response (Lenton, 2013). Palaeorecords across the PETM indicate that an abrupt release of isotopically-light carbon (between 2000 and 13000 Pg C, best estimate ~3000 Pg C) into the ocean-atmosphere system occurred in under ~5 kyr, accompanied by global warming of ~5 °C, a 2.5 to 3.0 ‰ benthic $\delta^{13}C$ excursion, and significant ocean acidification (Dickens, 2011; Dickens et al., 1995; Dunkley Jones et al., 2013;

Frieling et al., 2017; Kirtland Turner et al., 2017; Littler et al., 2014; McInerney and Wing, 2011; Sluijs et al., 2007b; Zachos et al., 2005, 2008, Zeebe et al., 2009, 2016). It has been hypothesised that gradual warming during the late Palaeocene (Figure 1) eventually crossed a tipping point, either through an internal process or an external perturbation such as volcanism (Svensen et al., 2004), which triggered the extensive dissociation of a carbon cycle 'capacitor' such as methane hydrates in ocean sediments (Dickens, 2011; Dickens et al., 1995; Minshull et al., 2016), permafrost soil carbon (DeConto et al., 2012) or organic carbon from a source such as peat (Cui et al., 2011; Kurtz et al., 2003) that benthic $\delta^{13}C$ records and modelling indicate accumulated earlier in the Palaeocene (Dickens, 2011; Komar et al., 2013; Kurtz et al., 2003). This in turn led to a rapid increase in atmospheric $CO_2$ ($pCO_2$) and the subsequent amplification of global warming and carbon release in a positive feedback loop that shifted the Earth System to a warmer state for ~100 kyr. An alternative hypothesis is that a very large external perturbation of volcanic carbon caused the PETM (Gutjahr et al., 2017) with a smaller role for amplifying feedbacks within the carbon cycle and therefore no significant role for a tipping point.

The PETM was followed by the Early Eocene Climatic Optimum (EECO; Figure 1) containing subsequent hyperthermal events such as Eocene Thermal Maximum 2 (ETM2) at ~54 Ma and ETM3 at ~53 Ma, which are potentially paced by orbital eccentricity forcing interacting with long-term warming and discharging methane hydrate deposits to produce threshold responses past repeated tipping points (Archer et al., 2009; Kirtland Turner et al., 2014; Komar et al., 2013; Littler et al., 2014; Lourens et al., 2005; Lunt et al., 2011; Stap et al., 2010; Westerhold et al., 2007; Westerhold and Rohl, 2009). However, the PETM occurred in a different orbital setting to the later events, suggesting that the PETM required an additional external "push" while the latter hyperthermals were eccentricity-paced tipping points (Littler et al., 2014). This push could have come from the emissions of the contemporaneous North Atlantic Volcanic Province (NAVP) eruptions both before and during the PETM (Frieling et al., 2016; Gutjahr et al., 2017; Storey et al., 2007; Svensen et al., 2004). Methane release from hydrate dissociation may also have been significantly limited or delayed by sediment transport processes, potentially limiting its role as a positive feedback (Minshull et al., 2016).

Many complex systems have been found to include tipping points, thresholds beyond which even small changes in condition can trigger the system to abruptly transition into a new equilibrium state (Dakos et al., 2015; Held and Kleinen, 2004; Lenton, 2013; Lenton et al., 2008; van Nes et al., 2016; Scheffer et al., 2001, 2009). Theory suggests that, prior to reaching such a tipping point, a system will exhibit 'critical slowing down' of its dynamics – meaning a slowing recovery rate in response to perturbations – which can be detected as increasing trends in autocorrelation and variability in time-series data (Carpenter and Brock, 2006; Dakos et al., 2008, 2012a; Kéfi et al., 2013; Lenton, 2011; Lenton et al., 2012a; Scheffer et al., 2009). Changes in skewness and kurtosis may also occur (although skewness can both increase or decrease depending on the nature of the alternative stable state and potential wells), and if internal variability is high, a system can 'flicker' between different states before undergoing a more permanent shift between them (Dakos et al., 2012a, 2013; Scheffer et al., 2009; Wang et al., 2012). Previous work has suggested that these indicators, which can be used as 'early warning signals' (EWS) or more generally as

metrics of resilience (the capacity of a system to recover from disturbance and return to its reference state (Grimm and Wissel, 1997; Holling, 1973; Scheffer et al., 2001)), may be detectable prior to some abrupt climate transitions in the palaeorecord (Dakos et al., 2008; Lenton, 2011), including the Eocene-Oligocene Transition and during several Pleistocene climate shifts (Dakos et al., 2008; Lenton, 2011; Lenton et al., 2012b, 2012a). However, autocorrelation and variance can also increase prior to non-catastrophic transitions, with or without bifurcations in phase space (Kéfi et al., 2013), or under some specific circumstances even decrease (Boettiger and Hastings, 2012b; Dakos et al., 2012b; Livina et al., 2012). Hence here increasing autocorrelation and variance is viewed more broadly as indicating declining resilience of a system (i.e. weakening negative feedbacks and greater sensitivity to small shocks), whether or not a critical transition is imminent. Other potential issues with detecting changing system resilience in palaeorecords include infrequent sampling rate, dating uncertainties, the possibility of producing false positives or negatives, and the extent to which these methods are dependent on subjective parameter choices (Boettiger et al., 2013; Boettiger and Hastings, 2012a; Lenton, 2011) (see Methods for further discussion).

Here we test the hypothesis that the PETM and ETM2 are examples of tipping points being reached in the carbon-climate system following long-term destabilisation (e.g. of a sensitive carbon cycle capacitor rich in isotopically-light carbon), by looking for declining resilience preceding them using published methodologies (Dakos et al., 2008, 2012a, Lenton et al., 2012a, 2012b). Whilst a signal of declining resilience cannot prove the existence of a tipping point, its absence would tend to falsify the tipping point hypothesis. Palaeorecords suffer from greater dating uncertainties and a less frequent sampling rate than is possible with modern climate data, making robust time-series analysis more challenging. Hence sufficiently long and high-resolution palaeorecords available across the late Palaeocene and early Eocene were required in order to enable significant results to be obtained. To this end we use the ~7.7 Myr-long benthic $\delta^{18}O$ and $\delta^{13}C$ palaeorecords from ODP Site 1262 in the South Atlantic (Littler et al., 2014), and sub-divide the datasets into pre-PETM and post-PETM bins, as well as sub-dividing the post-PETM bin into pre-ETM2 and post-ETM2 bins, for separate analyses. These isotope records track the long-term global state of high latitude climate and the carbon cycle respectively (Zachos et al., 2001, 2008) and are therefore appropriate data choices for detecting the resilience of the global carbon-climate system, which in turn determines the long-term resilience of the whole Earth System as its key slow-timescale components.

A major limitation of the available palaeorecords is that their resolution is of the order of ~3 kyr, which only allows us to monitor changes in the dynamics of the slowest parts of the carbon cycle and climate system (i.e. >10 kyr). For the carbon cycle these could include the silicate weathering feedback, which is hypothesised to act as the main long-term negative feedback on atmospheric $CO_2$ and therefore climate change (Berner, 1991; Berner et al., 1983; Kump and Arthur, 1997; Urey, 1952; Walker et al., 1981; Walker and Kasting, 1992), the strength of the biological pump and carbon burial rates in the ocean (Boscolo-Galazzo et al., 2018; Chamberlin, 1898; Derry and France-Lanord, 1996; France-Lanord and Derry, 1997; Hay, 1985; John et al., 2014), and medium-term fluctuations in the storage capacity of carbon reservoirs such as the deep ocean, methane hydrates, permafrost, or soil carbon (Batjes, 1996; Buffett and Archer, 2004; Cui et al., 2011; DeConto et al., 2012;

Dickens et al., 1995; Klinger et al., 1996; Tarnocai et al., 2009). For the climate system, slow processes could include substantial reorganisations of ocean circulation (Hofmann and Rahmstorf, 2009; Rahmstorf, 2002; Stocker and Wright, 1991; Stommel, 1961) and the growth or collapse of large ice sheets (although no substantial ice sheets existed at this time) (DeConto et al., 2008; DeConto and Pollard, 2003; Pagani et al., 2011; Pollard and DeConto, 2009). Any shorter-term drivers of instability closer to the event, for example changes in ocean and atmospheric dynamics or precursor warming on millennial timescales (Secord et al., 2010; Sluijs et al., 2007a), will be missed and thus could constitute 'missed alarms'. As a result, in this study we focus only on the long-term processes in the global carbon-climate system prior to and across the PETM and ETM2.

We use multiple indicators – including autoregressive coefficient at lag 1 (AR(1)) and detrended fluctuation analysis h-value (DFA-h) (Lenton et al., 2012b; Livina and Lenton, 2007) to reveal slowing down, and standard deviation (SD) and non-parametric drift-diffusion-jump (DDJ) model function metrics (Dakos et al., 2012a) to reveal increasing variability. An overall increasing trend in AR(1) or DFA-h would show the slow parts of the climate or carbon systems were recovering more slowly from regular perturbations, while increasing SD or variance as measured by the DDJ model would show each system was being perturbed further from their current state. Together they indicate a system being destabilised and becoming less resilient to being knocked into a new state. Skewness and kurtosis are also measured to provide further context (see Supplementary Material) as both may change in the presence of more extreme values. Sensitivity analyses are conducted in order to ensure detected signals are robust across different methodologies and parameter choices (see Methods and Supplementary Material).

## 2.    Methods

### 2.1.    Rolling Window Metrics

For the rolling window metrics we follow the methodology first outlined in a climate context by (Held and Kleinen, 2004), and subsequently used by other studies including (Dakos et al., 2008, 2012a, Lenton et al., 2012a, 2012b; Livina and Lenton, 2007), and the 'Early Warning Signals Toolbox' developed based on this work (documented at www.early-warning-signals.org and available in as the 'earlywarnings' package in R (R Foundation for Statistical Computing, 2016)). After selecting the dataset and for the pre-PETM analysis terminating it just prior to the hypothesised transition to avoid biasing the analysis, the data are first interpolated (using linear interpolation by default with the 'interp1' function in Matlab (The MathWorks Inc., 2016)) to provide the equidistant data-points required for rigorous statistical analysis and assumed by the AR(1) model. However, interpolation itself can introduce statistical artefacts into the analysis as, by definition, the addition of interpolated data-points increases self-similarity and thus autocorrelation in the dataset. In palaeorecords this tends to result in an artificial increase in autocorrelation in parts of the dataset with either sparser data-points or complete gaps in the data, but in this record there is no marked systematic shift in data time-steps (Supplementary Figure S5). As a result we also analyse non-interpolated data in order to assess the sensitivity of our results to interpolation. Following this, the data are then detrended by subtracting

the smoothed dataset, estimated with a Gaussian kernel smoothing function (using the '*ksmooth*' function in R), in order to remove any long-term trends as these are not the focus of the analysis. This makes the dataset stationary – a necessary prerequisite for time-series analysis – but this also somewhat reduces the value of lag 1 autocorrelation in the results. Bandwidth is an important consideration in this process and is adjusted heuristically for the datasets in order to best remove long-term trends but leaving short-term fluctuations, in this case giving a Gaussian kernel bandwidth of 0.1. This removes long-term secular trends and orbital cyclicity (>100 kyr) (Figure 2 & Figure 3, top panels), leaving only the short-term noise that reveals the resilience of the underlying longer-term (>10 kyr) processes.

An autoregressive model of order 1 (AR(1)) is fitted to the data within a rolling window (using the '*generic_ews*' function of the '*earlywarnings*' package in R). The AR(1) model is of the form: $x_{t+1} = \alpha_1 x_t + \varepsilon_t$, fitted by an ordinary least-squares method with a Gaussian random error and a constant time-step. Following previous studies the default window size is set at half the length of the dataset, but as part of our sensitivity testing we also repeat our analyses for window sizes between 25% and 75% (Supplementary Figures S1-S4). The choice of window length is a trade-off between dataset resolution and the reliability of the estimate of the indicator, with a short window allowing shorter-term changes in indicators to be tracked at the cost of lower estimate reliability and vice versa. On the same rolling window the absolute skewness, kurtosis, and standard deviation of the dataset are also calculated (also using the '*generic_ews*' function of the '*earlywarnings*' toolbox in R). Detrended fluctuation analysis h-value (DFA-h) was also used as an alternative measure to AR(1) for short-term memory and critical slowing down in the dataset bins (performed using the '*DFA*' function of the '*fractal*' package in R). DFA extracts the fluctuation function over a window $s$, and if the data is long-term power-law correlated, the fluctuation function $F(s)$ increases as a power law: $F(s) \propto s^h$, where $h$ is the DFA fluctuation exponent (Peng et al., 1994) and reaches value 1.5 at a critical transition (Lenton et al., 2012b; Livina and Lenton, 2007).

Finally, the likelihood of there being a real trend in the results is calculated by estimating the nonparametric Kendall rank-correlation statistic ($\tau$), which measures the strength of an indicator's tendency to increase (>>0) or decrease (<<0) against the null hypothesis of randomness (~0) (also using the '*generic_ews*' function of the '*earlywarnings*' toolbox in R). However, this statistic is most robust when the trend is consistent over a long period, while increasing but oscillating trends or trends only at the very end of the record can produce weak or even negative values despite a clearly visible trend (Dakos et al., 2012a). We calculate a p-value for each metric by bootstrapping our detrended datasets to generate 1000 surrogate records (or arima model-generated for AR(1) and DFA-h) with equivalent mean and variance, re-calculating the metric and Kendall $\tau$ value for each, and finding the proportion of Kendall $\tau$ values equal or greater than that of the original palaeorecord (Dakos et al., 2008).

## 2.2.    Binned metrics

As well as performing rolling window time-series analysis, we also measure AR(1), SD, skewness, and kurtosis on data (detrended but not interpolated) binned into pre- and post-event bins and excluding the events themselves to provide simple

before/after comparisons of changes across the events. To this end the datasets (n=2302) were binned into pre-PETM (n=1331), post-PETM (n=921), PETM to ETM2 (n=593), and post-ETM2 (n=240) bins, excluding data-points from during each event so as to avoid biasing by extreme or missing data. A p-value is calculated for each metric using a permutation test (i.e. by reshuffling and repartitioning the before/after event data into the same sized bins 1000 times and comparing the metrics'

resultant before/after differences with the observed metric before/after difference), except for AR(1) and DFA-h for which we instead use AR(1) model-derived surrogate data to compare against (i.e. by generating 1000 surrogate datasets with the same AR(1) value, mean, and variance as the before-event bin over the length of the after-event bin and compare this distribution to the observed after-event AR(1) value).

## 2.3.    Nonparametric Drift-Diffusion-Jump Model

A model-based alternative to the time-series analysis methods (whether rolling window or bin-based) above is to fit a general nonparametric drift-diffusion-jump model to the dataset with as a surrogate for an unknown data-generating process (Carpenter and Brock, 2011; Cox and Ross, 1976; Dakos et al., 2012a; Johannes, 2004). In this model functions are estimated for drift, diffusion, and jump processes using nonparametric regression, where drift measures the local rate of change, diffusion measures the standard deviation of the relatively small shocks that occur at each time step, and jumps are large intermittent

shocks. The conditional variance of the data is also estimated from the nonparametric regression, and represents the variance of the data from its conditional expectation estimated using kernel regression. We use the '*ddjnonparam_ews*' function in the '*earlywarnings*' package on R, using the default options of a bandwidth of 0.6 and 500 points for computing the kernel. We use raw data for this analysis, with no detrending or interpolation and without log transforming the data first. In interpreting the results we focus on the general long-term trends in the estimated terms as many of the shorter-term fluctuations potentially

represent model over-fitting.

## 2.4.    Limitations

Despite positive results in some palaeoclimate EWS studies (Lenton, 2011), there are several potential issues with searching for resilience indicators in palaeorecords. Palaeorecords suffer from greater dating uncertainties and a less frequent sampling rate than is possible with modern climate data, making robust time-series analysis more challenging. Most indicators also do

not reveal exact information about the nature of the transition itself, with increasing slowing down and variability detected prior to both catastrophic and non-catastrophic transitions featuring a bifurcation in phase space and even before non-catastrophic transitions without a bifurcation (Kéfi et al., 2013). Concerns have also been raised over the likelihood of producing false positives (where EWS appear to indicate an impending transition which never occurs) or false negatives (i.e. a "missed alarm", when EWS may be entirely absent prior to a known critical transition), and the extent to which these methods

are dependent on subjective parameter choices (Boettiger et al., 2013; Lenton, 2011). There is a risk that selecting and analysing known or suspected critical transitions in the palaeoclimate record is particularly liable to false positives, as positive indicators at the transitions could potentially have occurred purely by chance rather than due to systemic instability (Boettiger and

Hastings, 2012a). However, it has been argued that EWS can be reliably detected if both increasing autocorrelation and variance are seen prior to the transition rather than one of these indicators alone (Ditlevsen and Johnsen, 2010). Detecting multiple, consistent, and robust signals from the indicators can be indicative of decreasing system resilience even if a catastrophic transition is not reached or is instead triggered by an external perturbation rather than internal processes (Dakos et al., 2015).

## 3. Results and Discussion

### 3.1. Rolling Window Metrics

Rolling window metrics prior to the PETM reveal a sudden increase in AR(1) and SD after ~58.2 Ma in the interpolated benthic $\delta^{18}$O record associated with a step in the data (and the benthic $\delta^{13}$C record peaking), which despite a temporary drop in DFA-h suggests some degree of destabilisation of the slow climate system prior to the PETM (Figure 2 & Supporting Figure S1). However, the subsequent decline of standard deviation after ~57 Ma (likely to partially be the result of earlier extreme data-points from ~58.5-59.5 Ma leaving the rolling window) does not support a tipping point involving deep ocean temperature at the PETM, which is also indicated by the non-significant bootstrapped p-values for the metrics of both interpolated and non-interpolated data. The non-interpolated DFA-h metric does show a significant increase (p=0.015), which would suggest systemic slowing down, but this does not match the non-interpolated AR(1) metric. Alternatively, these results could represent a 'missed alarm' as the shorter-term climate dynamics that might be critical to the dynamics of the tipping point are not sufficiently resolved by the available data.

The benthic $\delta^{13}$C record shows clearer evidence of declining resilience in the slow components of the carbon cycle, with long-term increases in AR(1), DFA-h, and SD in the run-up to the PETM with steps at ~58.2 Ma and ~57.3 Ma which are consistent across the sensitivity analyses (absolute skewness also increases, while kurtosis declines up to the PETM; Supporting Figure S1). Bootstrapped p-values indicate that the $\delta^{13}$C SD trend is significant (p=0.002) for the interpolated data while the $\delta^{13}$C AR(1) and DFA-h trends are significant (p=0.024 & p=0.021 respectively) for the non-interpolated data. This supports a long-term slowing down in benthic $\delta^{13}$C in the late Palaeocene, which may reflect a gradually-forced destabilisation of the global carbon cycle prior to the PETM.

Rolling window analysis across the whole of the late Palaeocene / early Eocene (LPEE) interval suggest but cannot prove systemic changes in carbon cycle and climate (in)stability across both the PETM and ETM2 (Figure 3 & Supporting Figure S2). Between the PETM and ETM2 $\delta^{18}$O AR(1) and DFA-h increase up until ~200 kyr before ETM2 and SD experiences a small temporary increase followed by a larger decrease. In contrast, all metrics for $\delta^{13}$C experience a rapid jump during the PETM and then remain relatively stable until ETM2. Following ETM2, $\delta^{18}$O AR(1) and DFA-h increase significantly while SD increases slightly, whereas for $\delta^{13}$C all metrics (as well as absolute skewness and kurtosis; Supporting Figure S2)

consistently increase. However, the bootstrapped p-values indicate that none of these trends are significant for the interpolated data, but that for the non-interpolated data the increase in AR(1) and DFA-h for both $\delta^{18}O$ and $\delta^{13}C$ are highly significant (p=0 to 0.001). This indicates that there is some evidence for slowing down – but not for increased variability – in both the $\delta^{18}O$ and $\delta^{13}C$ data and therefore the long-term climate system and carbon cycle across the LPEE interval into the Eocene, but this is

dependent on not interpolating the data prior to the analysis. It should also be recognised that the abrupt shifts in $\delta^{13}C$ at the PETM and ETM2 are not fully removed by detrending prior to the analysis, hence they are at least partly responsible for the upward steps in the indicators at the events.

## 3.2.    Binned Metrics

To address the issue of large excursions failing to be removed by detrending for the rolling window metrics, as well as the
issue of data gaps caused by dissolution at the peak of each event (Littler et al., 2014), we calculate aggregate metrics (i.e. no rolling windows) on the binned data (excluding data from within the events) (Table 1). The binned metrics show significant increases in AR(1) across both the PETM and ETM2 for $\delta^{18}O$ and $\delta^{13}C$, with DFA-h also significantly increasing across the PETM and ETM2 for $\delta^{18}O$ and across ETM2 for $\delta^{13}C$. SD significantly falls in the interval between the PETM and ETM2 and then significantly increases across ETM2 for both datasets, but the significant increase for $\delta^{13}C$ for the all data after the PETM
is likely to be biased by the inclusion of extreme data-points within ETM2. These results are consistent with the rolling window metrics, except for $\delta^{13}C$ SD which in the rolling window metrics is higher between PETM and ETM2 than before the PETM, and is likely to be the result of excluding the extreme data values during the PETM itself in the binned analysis (with SD beginning to drop just before the ETM2 in the rolling window analysis once the PETM leaves the window). These results indicate both the long-term climate system and carbon cycle slowed down to some extent after both the PETM and ETM2 (but
became less variable following the PETM itself until ETM2), providing support for the slow parts of global carbon-climate system being progressively destabilised through the LPEE interval and into the Eocene by both of the hyperthermal events but not for any tipping points.

## 3.3.    Nonparametric Drift-Diffusion Jump Model

Fitting a non-parametric drift-diffusion-jump model to the datasets provides independent model-based metrics to compare to
the rolling window metrics, with terms for the conditional variance measuring variance from dataset's conditional mean (estimated by kernel regression), diffusion measuring the standard deviation of regular small shocks at every time-step, and jump intensity measuring either irregular large shocks or flickering (Carpenter and Brock, 2011; Dakos et al., 2012a) (Figure 4). For benthic $\delta^{18}O$ this model reveals an overall increase in conditional variance and diffusion and a reduction in jump intensity ~2 Myr before the PETM, followed by a slight decrease in conditional variance after the PETM and intermittent
spikes in jump intensity and conditional variance during and following ETM2. This suggests the climate shifted to a state with higher variability featuring regular small shocks ~2 Myr prior to the PETM, became slightly less variable following the PETM, but featured larger irregular shocks during and after ETM2. While this suggests some degree of climate instability in the ~2

Myr before the PETM and following ETM2, there is no evidence of a critical transition in the climate system at the PETM or ETM2 themselves. In contrast, the benthic $\delta^{13}$C model reveals decreasing diffusion and increasing conditional variance and jump intensity in the 1.5 Myr run-up to the PETM, indicating increasing total variability driven by large irregular shocks and consistent with a critical transition being approached in the carbon cycle at the PETM (Dakos et al., 2012a). Conditional variance and jump intensity remain high and diffusion remains low for ~1 Myr after the PETM, before reversing ~300 kyr before ETM2 except for brief spikes in both diffusion and jump intensity during and after ETM2. This indicates that variability in the carbon cycle remained high and driven by large shocks for ~1 Myr after the PETM, but that variability mostly shifted towards smaller regular shocks prior to and after ETM2. This difference in the sources of variance prior to each event (the PETM is preceded by elevated jump intensity and overall conditional variance, whereas ETM2 is preceded by increased diffusion and decreased jump intensity) suggests potentially differing carbon cycle dynamics and drivers prior to each event. The shift in variability before ETM2 also slightly precedes the biotic turnover detected in both marine and terrestrial records in the ~200 kyr prior to ETM2, despite there being no obvious shift in the palaeorecords that may have driven this turnover (Westerhold et al., 2018). Overall the $\delta^{13}$C DDJ results are consistent with elevated carbon cycle instability following the PETM, but suggests that ETM2 was not preceded by the same dynamics as the PETM.

## 4.    Conclusion

In summary, both rolling-window metrics before and across the PETM, binned metrics, and nonparametric drift-diffusion-jump models indicate that there was a decline in variability – suggesting a loss of resilience – in the slow components of the carbon cycle before the PETM. Following both the PETM and ETM2 there is also evidence of slowing down in both the long-term carbon cycle and climate system, indicating that both events led to a longstanding destabilisation of the carbon-climate system. In contrast, while there is some evidence for destabilisation in the $\delta^{18}$O data prior to and after the PETM, there is no clear evidence of a critical transition in the climate system at this time. Minimal lag between $\delta^{18}$O and $\delta^{13}$C in the late Palaeocene indicates close coupling between climate and the carbon cycle prior to the PETM (Littler et al., 2014), and so the observed instability in the climate system is likely to have been induced by the contemporaneous destabilisation of the carbon cycle. Furthermore, ETM2 appears to be preceded by different carbon cycle dynamics to the PETM, which fits with the suggestion that the PETM required an extra "push" unlike the later eccentricity-paced hyperthermals which might represent more classical tipping points (Littler et al., 2014). These results are consistent with the hypothesis of a gradual destabilisation of the long-term carbon cycle in the ~1.5 Myr preceding the PETM (starting at ~57.5 Ma, intensifying after ~56.5 Ma) associated with increasing $pCO_2$ concentrations and the warming seen in the benthic $\delta^{18}$O record from ~58.2 Ma. This coincides with the North Atlantic Volcanic Province (NAVP) eruptions between 61 and 57 Ma and its associated volcanic and thermogenic $CO_2$ and methane emissions, with subsequent large-scale eruptions from ~56.1 Ma (possibly preceded by some degree of cryptic degassing (e.g. Armstrong McKay et al., 2014) coinciding with the intensified carbon cycle destabilisation from ~56.5 Ma) potentially triggering or prolonging the PETM (Frieling et al., 2016; Storey et al., 2007; Svensen et al., 2004).

This time also coincides with a dramatic long-term decrease in organic carbon storage following a large build-up as indicated by the downturn in benthic $\delta^{13}C$ from ~58 Ma (Figure 1), either as a result of large-scale methane hydrate or peat dissociation and oxidation in response to warming (Dickens, 2011; Komar et al., 2013; Kurtz et al., 2003) or a reduction in marine biological pump strength as higher temperatures lead to increased respiration rates of particulate organic carbon (Boscolo-Galazzo et al., 2018; John et al., 2014). This weakening of the organic carbon burial feedback in response to volcanism-driven warming could be the primary driver of the observed geological carbon cycle destabilisation in the ~1.5 Myr prior to the PETM, and may have in turn prolonged the duration of the PETM itself. Reconstructed silicate weathering feedback strength fell by ~40% in the ~3 Myr after the PETM due to reduced continental weatherability (Caves et al., 2016; van der Ploeg et al., 2018), potentially allowing the carbon-climate system to remain destabilised and susceptible to further shocks long after the PETM and ETM2 as observed. The hypothesis of a carbon cycle tipping point at the PETM survives our tests (although they cannot directly confirm it or rule out an external trigger). In contrast the hypothesis of a tipping point in deep ocean temperature (as recorded by the $\delta^{18}O$ record considered) is not supported. A large external perturbation, e.g. a massive, abrupt injection of volcanic carbon from the NAVP during the PETM (Gutjahr et al., 2017; Storey et al., 2007) or the suggestion of a meteorite strike (Schaller et al., 2016), could have played a role in triggering the PETM, but we find clear evidence that the carbon cycle had already been getting progressively more unstable and thus more vulnerable to being pushed beyond a tipping point, and remained so in its aftermath.

## Author Contributions

DIAM & TML designed the study; DIAM performed the analyses; DIAM & TML wrote the paper.

## Data availability

The data used for the plots and analysis of the global Palaeocene and Eocene benthic isotope stack (Zachos et al., 2001, 2008) and the ODP Site 1262 benthic isotopes (Littler et al., 2014, with ages adjusted as per Westerhold et al., 2015) are available via their published sources.

## Conflicts of Interest

The authors declare that they have no conflict of interest.

## Acknowledgements

This work was supported by an EPSRC/ReCoVER Early Career Research Project Award (Number: RFFECR 002) and a Natural Environment Research Council Studentship to DIAM (Number: NE/J500112/1). TML was supported by a Royal

Society Wolfson Research Merit Award. We are grateful to Kate Littler for the data used in this study and for discussing our results, and to Toby Tyrrell, Paul A. Wilson, and James G. Dyke for feedback on preliminary results, drafts, and interpretations.

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

**Figures**

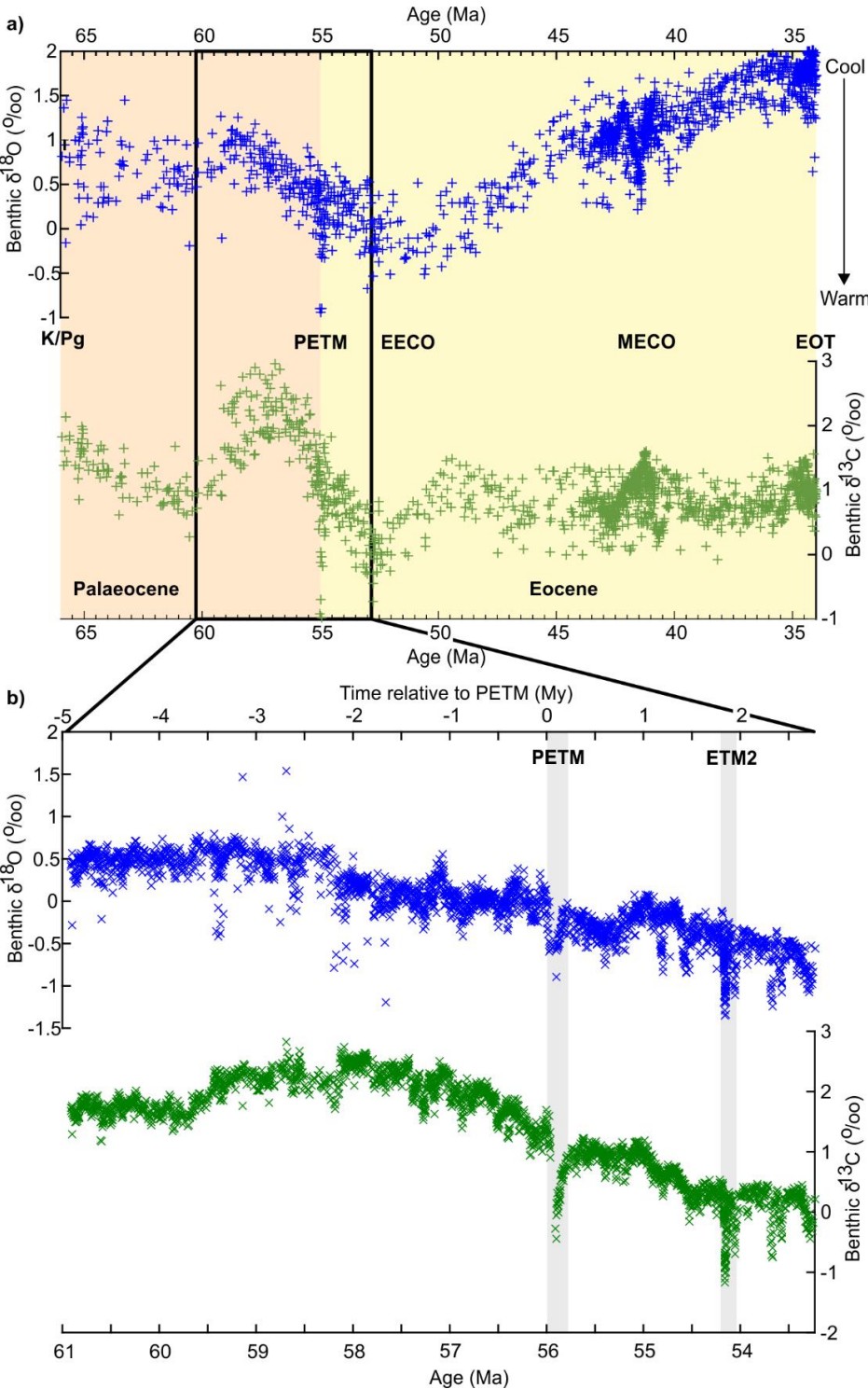

**Figure 1: Palaeorecords of benthic δ¹⁸O (blue) and δ¹³C (green) across a) the Palaeocene and Eocene (data from global stack (Zachos et al., 2001, 2008)) and b) the Late Palaeocene-Early Eocene (LPEE) study interval (data from ODP Site 1262 (Littler et al., 2014) with ages adjusted as per (Westerhold et al., 2015)).** Significant climate and carbon cycle events are labelled, including the Cretaceous/Palaeogene boundary (K/Pg), Palaeocene-Eocene Thermal Maximum (PETM), Eocene Thermal Maximum 2 (ETM2), Early Eocene Climatic Optimum (EECO), the Mid-Eocene Climatic Optimum (MECO), and the Eocene-Oligocene Transition (EOT), while the black box marks the LPEE interval analysed in this study. The mismatch in PETM date between the two datasets is a result of the updated age model of (Westerhold et al., 2015) applied to the record of (Littler et al., 2014) but not (Zachos et al., 2001, 2008), which we have maintained so as to maintain consistency with published records.

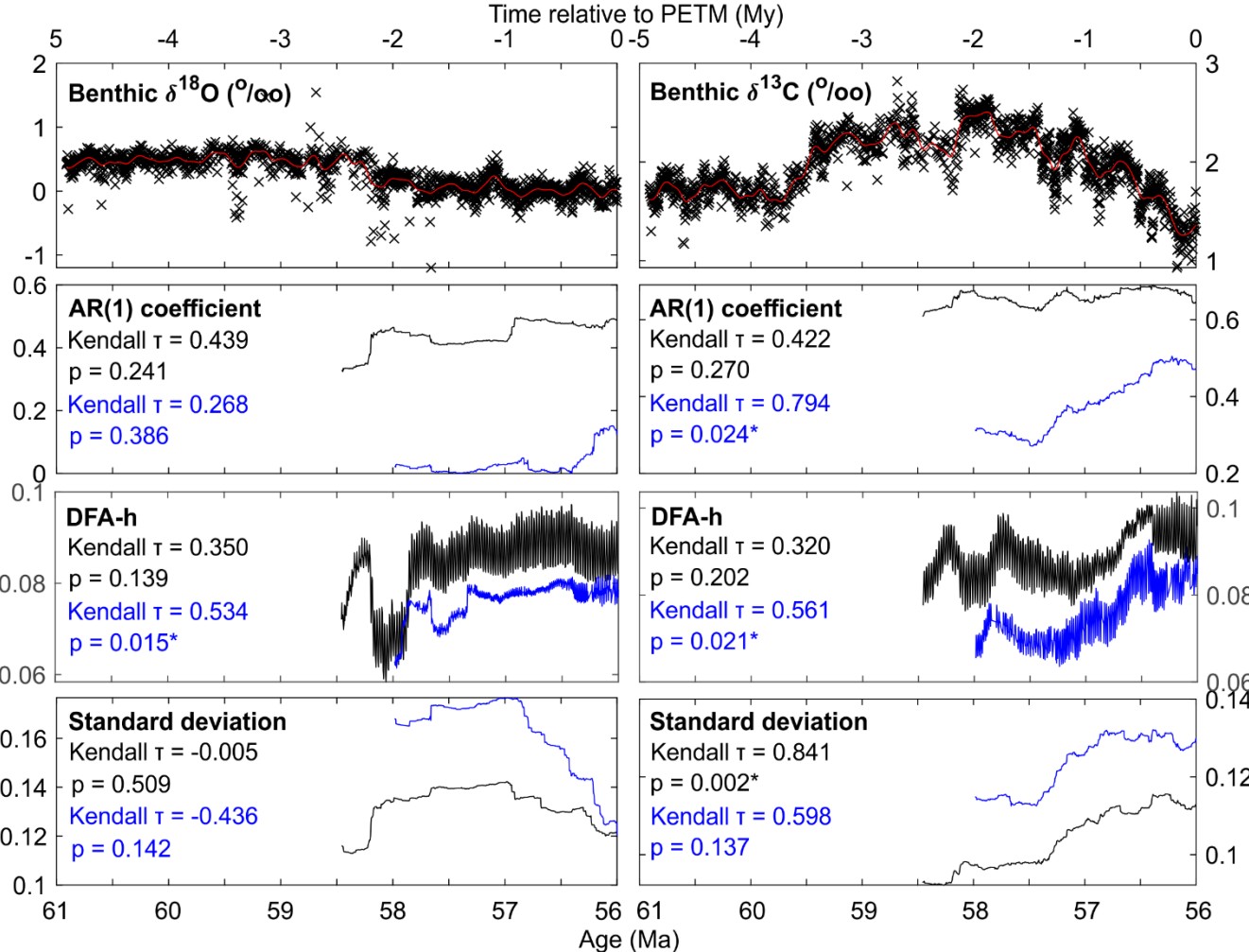

**Figure 2: Rolling window resilience analysis of benthic δ¹⁸O (left) and δ¹³C (right) in the run-up to the PETM.** The top panels illustrate the palaeorecord (black crosses) and the detrending applied to the data (red line), with the panels below illustrating the results of the analysis for AR(1) coefficient, detrended fluctuation analysis h-value, and standard deviation calculated in a 50% rolling window across each time-series for both interpolated (black line) and non-interpolated (blue line) data with the Kendall τ rank-correlation and boostrapped p-value for each. Results for skewness, kurtosis, and sensitivity analyses for all metrics can be found in the Supplementary Material.

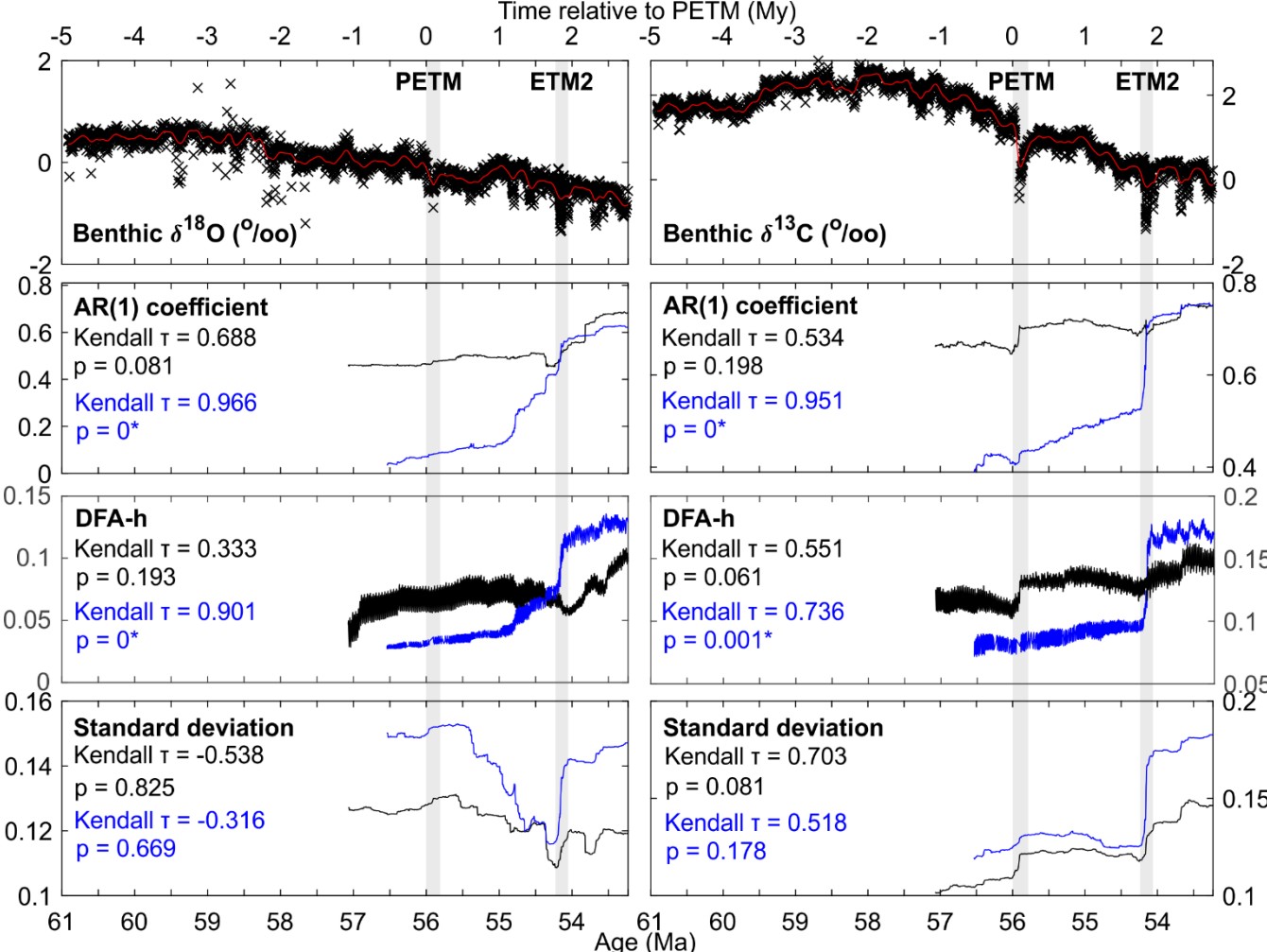

**Figure 3: Rolling window resilience analysis of benthic δ¹⁸O (left) and δ¹³C (right) across the PETM and ETM2.** The top panels illustrate the palaeorecords (black crosses) and the smoothed record used to detrending the data (red line), with the panels below illustrating the results of the analysis for AR(1) coefficient, detrended fluctuation analysis h-value, and standard deviation calculated in a 50% rolling window across each time-series for both interpolated (black line) and non-interpolated (blue line) data with the Kendall τ rank-correlation and boostrapped p-value for each. The PETM and ETM2 are marked by the grey bars. Results for skewness, kurtosis, and sensitivity analyses for all metrics can be found in the Supplementary Material.

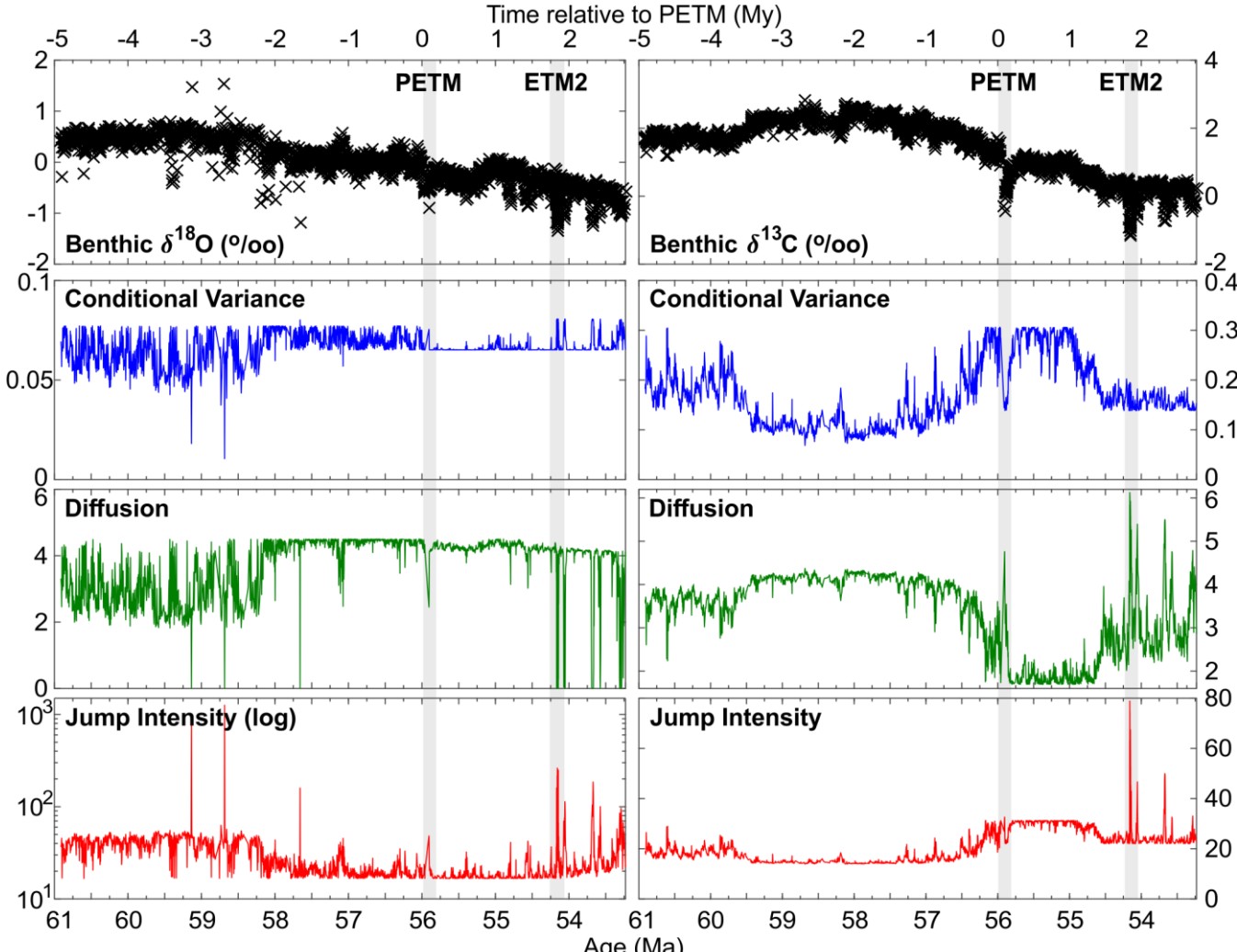

**Figure 4: Non-parametric drift-diffusion-jump model functions of benthic δ¹⁸O (left) and δ¹³C (right) across the PETM and ETM2.**
The top panels illustrate the palaeorecords (black crosses), with the panels below illustrating the model functions for conditional variance (blue line), diffusion (green line), and jump intensity (red line) for each palaeorecord (drift is not plotted). The PETM and ETM2 are marked by the grey bars.

## Tables

**Table 1: Values of binned metrics for both benthic δ¹⁸O (top) and δ¹³C (bottom).** Bins are of all data (detrended but not interpolated) before the PETM, all data after the PETM (including data from within ETM2), all data between the PETM and ETM2, and all data after ETM2. The indicators are AR(1) coefficient, detrended fluctuation analysis h-value (DFA-h), standard deviation (SD), skewness (SKEW), and kurtosis (KURT). Green shading indicates the indicator has increased relative to before the event (either the PETM or ETM2), red shading indicates a decrease in value. Each value is followed by a p-value (*in italics within parentheses preceded by a \* if significant*) computed using a permutation test, except for AR(1) for which we instead use AR(1) model-derived surrogate data to compare against (see Methods for details).

| Metric | Before PETM | After PETM (all) | After PETM (to ETM2) | After ETM2 |
|--------|-------------|------------------|----------------------|------------|
| **δ¹⁸O** | | | | |
| AR(1) | 0.0611 | 0.659 *(*0*) | 0.509 *(*0*) | 0.657 *(*0*) |
| DFA-h | 0.035 | 0.140 *(*0*) | 0.098 *(*0*) | 0.187 *(*0*) |
| SD | 0.146 | 0.152 (*0.279*) | 0.119 *(*0.991*) | 0.145 *(*0.002*) |
| SKEW | -1.447 | -0.635 *(*0.098*) | -0.927 (*0.281*) | -0.713 (*0.211*) |
| KURT | 14.621 | 4.905 *(*1*) | 4.490 *(*0.999*) | 3.884 (*0.741*) |
| **δ¹³C** | | | | |
| AR(1) | 0.402 | 0.784 *(*0*) | 0.500 *(*0.001*) | 0.743 *(*0*) |
| DFA-h | 0.071 | 0.189 *(*0*) | 0.088* (*0*) | 0.203 *(*0*) |
| SD | 0.122 | 0.189 *(*0*) | 0.105 *(*0.999*) | 0.187 *(*0*) |
| SKEW | -0.541 | -1.165 *(*0.98*) | -0.700 (*0.845*) | -1.056 (*0.828*) |
| KURT | 4.046 | 7.634 *(*0.005*) | 4.073 (*0.415*) | 4.728 (*0.307*) |