# Peer review of "Reduced Carbon Cycle Resilience across the Palaeocene-Eocene Thermal Maximum"

_Climate of the Past, 2018_

## Referee Comment (RC1) · V. N. Livina (Referee) · 9 Jul 2018

The paper by McKay and Lenton "Reduced carbon cycle resilience across the Paleoscene-Eocene thermal maximum" investigates paleo data for early warning signals prior to known paleo events and discusses the system resilience in geochemical context. The paper is well-written, applies previously developed and tested techniques, and can he published after a revision.

I suggest the following modifications.

The authors could add definitions and references to the ideas of tipping points and

resilience.

In Figure 1, in the upper panel the time series have local minima at 55Ma, whereas in the enlarged bottom panel these minima are located at 56Ma. Why?

The paper does not include the reference [Held and Kleinen, GRL 2004], which was the first to apply the lag-1 autocorrelation as an early warning indicator of a climatic bifurcation. The authors should cite it in page 2, line 25; in page 4, line 6, and discuss the H&K contribution as pioneers of this technique in climatology.

Variance does not necessarily increase prior to a bifurcation, see the counter-example in [Livina et al, Physica A 2012]. Changes in skewness and curtosis are often consequences of the asymmetric effects due to the appearing or disappearing potential wells. This can be discussed in more detail in page 2.

It is not clear to me why the authors separate the cases of rolling windows for indicators and binned indicators. As I understand it, the binned indicator is a particular case of a rolling window, when the shift of the window is equal to the size of the window. Why do authors stress that, for instance, DFA is a binned indicator? It can be estimated in rolling windows just the same. Accordingly, I do not understand the comment in brackets in page 5 , line 21, which separates rolling window and "metric-based".

When using several sizes of windows for an indicator (the figures in the supplement), the authors could do this not just for three sizes but for a range of windows, with estimation of uncertainty in the indicator curve - see the example of such indicator curves in [Livina et al, JCSHM 2013].

In page 4, line 6, "the methodology was first outlined in [Held and Kleinen, GRL 2004], then used in [Livina and Lenton, GRL 2007]", etc.

In page 4, lines 17-18: while trends may be not the focus of the analysis, their removal reduces the value of the lag-1 autocorrelation, and this should be kept in mind in analysing the indicators.

For the DFA reference in page 5, line 14, please use instead of [Lenton et al 2012b] the original DFA reference [Peng et al, PRE 1994].

In page 1, line 17, what do the authors mean by "differing"?

In page 8, lines 2-3, to avoid line break between 2 and My, use LaTeX command $\sim$

Valerie Livina

---

## Referee Comment (RC2) · P. Maffre (Referee) · 12 Jul 2018

First, I have to admit that this study uses statistical methods for time-series analysis which I am not very familiar with. So my remarks may be naive. Nevertheless, care has been taken to explain how to interpret the results given by these methods, as well as the limitations in the eponymous section. This is appreciable for a unfamiliar reader.

As far as I know, this methodology of early warning signals for detecting tipping points is relatively new in paleoclimate studies. It makes this study all the more interesting and the results worth to be broadly communicated. The use of different indicators (distribution moments, autoregressive model, detrended fluctuation analysis, drif-diffusion-jump

model) and the sensitivity analysis contribute to strengthen this study. Moreover, this methodology has not been applied to paleorecords earlier than Quaternary according to the cited literature (Dakos et al., 2008, Lenton, 2011, Lenton et al., 2012a, 2012b). Despite the difficulties of using data as far in the past of the Earth —that are clearly mentioned in the manuscript— this study constitutes therefore a significant advance. I would suggest that this pioneering application should be highlighted in the main text.

Here follows my comments.

General remarks:

This study deals with the reduction of climate and carbon cycle resilience with very little mention of the processes responsible for this (lack of) resilience. It is acknowledge Page 6 lines 4-6 that "most indicators do not reveal exact information about the nature of the transition itself" and I understand it is not the aim of the study. Nonetheless, for periods of time as extended as the one covered by this study (5 and 8 million years, which is substantially more than the previous EWS studies mentioned), it is commonly hypothesized than silicate weathering is the feedback that stabilize Earth climate (Walker, Hays and Kasting (1981) J Geophys Res 86, 9776; Berner and Caldeira (1997), Geol, 25, 955; François and Goddéris (1998), Chem Geol, 145, 177-212). The authors should explicitly say if they aim at tracking carbon-climate resilience due to silicate weathering feedback (or more generally feedbacks in geological carbon cycle), or only "shorter-term" climate resilience (for instance, the processes mentioned Page 3, lines 21–23), or both. It is of particular importance because of the timescale of considered perturbations of d13C and d18O: It is specified Page 4, lines 16-19 that the data are detrended in order to remove any long-term trends. This is indeed essential to get stationary time-series. But though it is explicitly said that bandwidth "is an important consideration", the only given information is that it is "adjusted heuristically for each dataset". More precision should be given on bandwidth value (is it constant along one given time-serie?) and above all because if it is less than the response time of carbon cycle (∼100–200ky, François and Goddéris, 1998), then the indicators are not

(or only partially) measuring the resilience due to its feedbacks. From the timescale of variations of smoothed records shown in Fig. 3, 4 S1 and S2 (red lines), I guess the bandwidth is actually in the order of 100ky. The choice of the timescale of the fluctuations to study and the feedbacks to investigate is up to the authors, but it should be specified, and conclusions can be drawn only for the focused feedbacks.

The section 3.2 (Binned Metrics) and Table 1 show that most of the indicators exhibit significant variations before/after the PETM and the ETM2. This suggest (as mentioned Page 7, lines 26–28) that the hyperthermal events are partly responsible for the loss of resilience of the carbon-climate system, or at least that these events are not simple perturbations followed by a relaxation towards the same "initial state": they induce some permanent or irreversible changes. Even if evidences of tipping points are lacking (as said Page 7, lines 27–28), I think it is an interesting results and should be highlighted in the conclusion, where there is no mention of this fact. I also wonder which component of carbon-climate system can be expected to undergo irreversible changes. If there is any "good candidate", it may be interesting to precise it.

Minor specific remarks:

Page 4, line 15 and line 22: With the definition of autoregressive model as 'x(t+dt) = a*x(t) + e(t)', a constant timestep 'dt' is inherently necessary for the autoregressive coefficient 'a' is directly link to 'dt'. I wonder then how to fit an autoregressive model to the non-interpolated data? Just dividing 'a' by the local timestep is enough? It is not precised in the description of the 'generic_ews' function in R. Perhaps it is a "routine analysis" and is not worth to be precised, I can't really judge it.

Page 6, line 21: The word "divergence" (of standard deviation) may be misleading because the reader would firstly expect to find an "increase", which is in contradiction with the rest of the sentence (and the Figure). Perhaps it is preferable to substitute it for "decrease of standard deviation" or "reducing standard deviation". In addition, this decrease of SD is likely to be due to earlier "extreme" events (between 59.5 and

58.5Ma) that slip out of the rolling window when it reaches 57-56Ma. Indeed, with a 75% rolling window (Fig. S1), SD does not exhibit such a decrease. It does with a 25% rolling window probably because there are "extreme" events up to 57.5Ma. Therefore, this decrease of SD could be not linked to what happen immediately before the PETM, and not contradicting with the increase of AR1.

Page 6, line 29: How to interpret a decrease in kurtosis? Shouldn't we except while approaching a tipping point more frequent extreme deviations, and then a higher kurtosis?

Page 7, lines 10–11: "but that the increase of AR(1) for both d18O and d13C are highly significant (p=0)". Please add "for non-interpolated data" to be explicit.

Page 7, lines 13-15: There is another evidence that the values of d13C during the events (at least for the PETM) are partly responsible for the upward steps in the indicators: with a 25% rolling window (Fig. S2), both AR1 and SD show a downward step at 54Ma, exactly when the PETM leaves the rolling window (and after the ETM2 has come into the rolling window). However, it is true that the binned metrics clearly show steps that are not due to what happen during the events.

Page 8, lines 2-3: Same remark as for the decrease of SD before PETM: the decrease of jump intensity can be due to the "extreme" events between 59.5Ma and 57.5Ma.

Supplementary Figures 1 and 2: How come than skewness is systematically positive? At least during PETM and ETM2, it seems there are more points beneath the red line than above. Moreover, in the binned metrics analysis (Table 1), skewness in systematically negative.

---

## Author Comment (AC1) · 26 Jul 2018

Thank you Valerie for a thoughtful and clear review of our paper. Here we will respond in brief to your comments and describe how we will subsequently revise the paper.

We indeed should have cited Held & Kleinen (2004), and will include this and other additional references – regarding tipping points and resilience in general as well as specifically on the alternative drivers of indicator behaviour (such as for skewness or kurtosis) and Peng et al (1994) for DFA – in the revised manuscript.

The difference in dates in Figure 1 are the result of differing age models between

the global Cenozoic isotope compilation (Zachos et al, 2008) and the more recent and localised high-resolution record with an updated age model from Westerhold et al (2015). We have left them as is to maintain consistency with the published records, but will mention and explain this difference in the revised manuscript in order to avoid confusion.

The binned metrics are indeed a special case of the rolling window, although with differing sized bins for pre-PETM, PETM-ETM2, and post-ETM2. The key aspect is that in contrast to the rolling window run over the whole dataset for the binned analysis the data from the events themselves are removed and so not included in these bins, avoiding the indicators being biased by the events themselves. However, we agree that the way this section is currently phased is a bit confusing, so we will rephrase this section and the comment on page 5 line 21 in the revised manuscript to make this clearer.

DFA could also be done on a rolling window along with the other EWS indicators. This was originally excluded due to time constraints in the initial project (with AR1 used as the main 'memory' indicator and DFA added later to just the binned section), but we can run and include this additional analysis in the revised manuscript. A continuous range of rolling window length rather than just a comparison of 25 / 50 / 75% windows would also be advantageous, and so we will also attempt a more robust sensitivity analysis of rolling window size.

In the abstract (page 1 line 17) by "differing carbon cycle dynamics preceding the PETM and ETM2" we refer to the drivers of variance being different for each event, with an increase in jump intensity (and overall conditional variance) in the d13C prior to the PETM versus an increase in diffusion (and a decrease in jump intensity) prior to ETM2. This difference suggests potentially differing carbon cycle dynamics prior to each event, which we discuss in Section 3.3. However, this could be phrased more clearly in the abstract and so will clarify this line (along with Section3.3) in the revised manuscript.

---

## Author Comment (AC2) · 26 Jul 2018

Thank you Pierre for a thoughtful and clear review of our paper. Here we will respond in brief to your comments and describe how we will subsequently revise the paper.

You are right to say that it is important to be clearer about which long-term carbon cycle or climate processes may be implicated in our analysis, and the silicate weathering feedback is indeed the most important of these for the geological carbon cycle (along with long-term changes in the burial rates of organic carbon and ocean carbonate). As mentioned in page 3 lines 20-26, due to the temporal resolution of our data we cannot resolve short-term processes (i.e. that take place over less then ~10 kyr), but in our

revised manuscript we will make this section clearer and be more explicit about which processes of the geological carbon cycle and long-term climate system we expect to play a role.

You are also right that bandwidth choice is important in the context of process timescales as well, and that the original manuscript failed to mention the bandwidth choices for the main rolling window metrics – we found that the optimal Gaussian kernel bandwidth was 0.1 and we will make this clear in the revised manuscript. This does not directly translate to a frequency limit, but as shown by the red line in Figures 2 and 3 this removes all of the secular trends and the long-term orbital-scale ($\sim$100+ kyr) variability. While this at first inspection would seem like we're filtering out what we're interested in (the long-term carbon cycle / climate system), with this methodology it is in fact the short-term noise that we are interested in as this short-term noise reveals the resilience of the longer-term processes. But as mentioned above, the data's temporal resolution places a lower limit on what timescale processes this method can reveal (in this case anything shorter than 10s of kyr). As a result, although we filter out the direct signal of longer-term processes, with this method this does not exclude these processes from affecting the results and interpretation.

Our results do indeed clearly show increased and persistent long-term destabilisation following both hyperthermal events in our record, and this has implications for the early Eocene (as a destabilised carbon-climate system may have played a role in the subsequent repeated hyperthermals). In our original manuscript we mostly focused on the implications of destabilisation for the presence of tipping points preceding the two hyperthermal events in our record, but you are right to point out that the subsequent persistence of the loss of resilience is important as well and in the revised manuscript we will make this clearer (as well as the processes likely to be implicated in this persistent loss of resilience, such as a dampened silicate weathering feedback).

Regarding the minor remarks, we will provide clarifications in the revised manuscript where requested. In brief, for the AR model time-step the R function simply assumes

a constant time-step throughout the time-series (i.e. it takes the time-series length and divides by N to get the constant dt). Of course without interpolation the actual data points will slightly differ from this dt which will introduce some error (and is why the interpolated run is considered the default, despite the alternative problem discussed in the text that this introduces instead), but there is no systematic bias in the distribution real data time-steps which limits any systematic error as a result. We will clarify that d18O SD declines just before the PETM and discuss why.

We also agree that one would expect kurtosis to increase prior to a tipping point as extreme data values become more common, but this is not considered to be as strong an indicator of resilience as other measures like variance or AR1. In this case it seems that the section in the d13C record between ∼58-59.5 Ma seems to have especially high kurtosis (as shown in supplementary fig 1 especially by the 25% window), and kurtosis gradually falls after this as this period leaves the rolling window, potentially masking any pre-PETM signal. The within-event data in the whole-dataset rolling window analysis does affect the results, which is why the excluding-event bins help to check the increases are there – we can add extra discussion of this in the revised manuscript. This isn't the case for the npDDJ model though, as this does not use a rolling window.

Lastly, for d13C skewness, there is indeed a mismatch between the skewness as calculated on a rolling window (>∼0 at all times and gradually increasing) and in the bins (<0 and decreasing for every bin). The events themselves should skew negative (as shown by the initial small drops in each in the rolling window analysis), yet the analysis including them is systematically positive while excluding them results in negative skewness. This is a consistent output despite using the same data, so it's not entirely clear why this is the case. We will further explore and then discuss the reasons behind this mismatch in the revised manuscript.

---

## Author Response (AR1)

**Authors' response to reviewers' comments on: "Reduced Carbon Cycle Resilience across the Palaeocene-Eocene Thermal Maximum" (Ms. Ref. No.: cp-2018-57)**

Dear Yannick Donnadieu and Reviewers,

Thank you for the timely reviews of our manuscript "Reduced Carbon Cycle Resilience across the Palaeocene-Eocene Thermal Maximum". Please find below our detailed responses to the reviews. We include the original comments and our response under each point in **bold** text and with line numbers referencing the revised manuscript with changes marked.

We hope we have sufficiently answered your queries in our response.

Yours sincerely,

David I. Armstrong McKay & Tim M. Lenton

**Response to Reviewer 1 (R1) Valerie Livina:**

The paper by McKay and Lenton "Reduced carbon cycle resilience across the Paleoscene-Eocene thermal maximum" investigates paleo data for early warning signals prior to known paleo events and discusses the system resilience in geochemical context. The paper is well-written, applies previously developed and tested techniques, and can he published after a revision.

I suggest the following modifications.

The authors could add definitions and references to the ideas of tipping points and resilience.

**The definitions and descriptions of tipping points and resilience have been clarified and additional references added (page 9 line 25 & page 10 line 1).**

In Figure 1, in the upper panel the time series have local minima at 55Ma, whereas in the enlarged bottom panel these minima are located at 56Ma. Why?

**The data in the top panel is from the Zachos compilation (2008), which uses an older age model for the data in this section, whereas the bottom panel uses higher-resolution data with an updated age model (Westerhold et al 2015) across just the study period. To maintain consistency with published records we have left them as-is, but we have clarified the difference in age models and PETM date in the caption.**

The paper does not include the reference [Held and Kleinen, GRL 2004], which was the first to apply the lag-1 autocorrelation as an early warning indicator of a climatic bifurcation. The authors should cite it in page 2, line 25; in page 4, line 6, and discuss the H&K contribution as pioneers of this technique in climatology.

**We indeed should have cited Held & Kleinen (2004), and have now included and discussed this reference where required (page 9 line 26, page 11 line 24)**

Variance does not necessarily increase prior to a bifurcation, see the counter-example in [Livina et al, Physica A 2012]. Changes in skewness and curtosis are often consequences of the asymmetric effects due to the appearing or disappearing potential wells. This can be discussed in more detail in page 2.

**We have added discussion of counter-examples to increasing variance, skewness, and kurtosis (page 9 line 31; page 10 line 6).**

It is not clear to me why the authors separate the cases of rolling windows for indicators and binned indicators. As I understand it, the binned indicator is a particular case of a rolling window, when the shift of the window is equal to the size of the window. Why do authors stress that, for instance, DFA is a binned indicator? It can be estimated in rolling windows just the same. Accordingly, I do not understand the comment in brackets in page 5 , line 21, which separates rolling window and "metric-based".

**The binned metrics are indeed a special case of the rolling window, although with differing sized bins for pre-PETM, PETM-ETM2, and post-ETM2. The key aspect is that in contrast to the rolling window run over the whole dataset for the binned analysis the data from the events themselves are removed and so not included**

**in these bins, avoiding the indicators being biased by the events themselves. However, we agree that the way this section is currently phased is a bit confusing, so we have rephrased this section to make it clearer (page 13 lines 3-6). The comment on page 5 line 21 has also been clarified (page 13 line 17).**

**You are also right to say that DFA can also be done on a rolling window along with the other EWS indicators.**
**This analysis was originally excluded due to time constraints in the initial project (with AR1 used as the main 'memory' indicator and DFA added later to just the binned section), but we have now run and included this additional analysis in the revised manuscript. These analyses strongly support our original AR1 results and interpretation, and discussion of these results have been included where necessary.**

When using several sizes of windows for an indicator (the figures in the supplement), the authors could do this
not just for three sizes but for a range of windows, with estimation of uncertainty in the indicator curve - see the example of such indicator curves in [Livina et al, JCSHM 2013].

**A continuous range of rolling window length rather than just a comparison of 25 / 50 / 75% windows is also advantageous, and so we have also included a more robust sensitivity analysis of rolling window size (presented as plots and histograms of kendall tau values for each 1% increase in rolling window sizes**
**between 25% and 75%) in the Supplementary Material (Supp. Figures S3 & S4) and mentioned in the text (page 12 line 16).**

In page 4, line 6, "the methodology was first outlined in [Held and Kleinen, GRL 2004], then used in [Livina and Lenton, GRL 2007]", etc.

**Clarification incorporated (page 11 line 24)**

In page 4, lines 17-18: while trends may be not the focus of the analysis, their removal reduces the value of the lag-1 autocorrelation, and this should be kept in mind in analysing the indicators.

**We have added a mention of this to the methods discussion (page 12 line 6)**

For the DFA reference in page 5, line 14, please use instead of [Lenton et al 2012b] the original DFA reference [Peng et al, PRE 1994].

**Citation incorporated (page 12 line 23)**

In page 1, line 17, what do the authors mean by "differing"?

**By "differing carbon cycle dynamics preceding the PETM and ETM2" we refer to the drivers of variance being different for each event, with an increase in jump intensity (and overall conditional variance) in the d13C prior to the PETM versus an increase in diffusion (and a decrease in jump intensity) prior to ETM2. This**
**difference suggests potentially differing carbon cycle dynamics prior to each event, which we discuss in Section 3.3. However, this could be phrased more clearly in the abstract and so will clarify this line (page 8 line 19) along with Section3.3 (page 16 line 18) in the revised manuscript.**

In page 8, lines 2-3, to avoid line break between 2 and My, use LaTeX command~

**This manuscript was composed in Microsoft Word, but we have adjusted this line to avoid the line break (page 16 line 7).**

**Response to Reviewer 2 (R2) Pierre Maffre:**

First, I have to admit that this study uses statistical methods for time-series analysis which I am not very familiar with. So my remarks may be naive. Nevertheless, care has been taken to explain how to interpret the results given by these methods, as well as the limitations in the eponymous section. This is appreciable for a unfamiliar reader.

As far as I know, this methodology of early warning signals for detecting tipping points is relatively new in
paleoclimate studies. It makes this study all the more interesting and the results worth to be broadly communicated. The use of different indicators (distribution moments, autoregressive model, detrended fluctuation analysis, drif-diffusion-jump model) and the sensitivity analysis contribute to strengthen this study. Moreover, this methodology has not been applied to paleorecords earlier than Quaternary according to the cited literature (Dakos et al., 2008, Lenton, 2011, Lenton et al., 2012a, 2012b). Despite the difficulties of using
data as far in the past of the Earth â˘ AˇTthat are clearly mentioned in the manuscriptâ˘ AˇT this study constitutes therefore a significant advance. I would suggest that this pioneering application should be highlighted in the main text.

Here follows my comments.

General remarks:

This study deals with the reduction of climate and carbon cycle resilience with very little mention of the processes responsible for this (lack of) resilience. It is acknowledge Page 6 lines 4-6 that "most indicators do not reveal exact information about the nature of the transition itself" and I understand it is not the aim of the study. Nonetheless, for periods of time as extended as the one covered by this study (5 and 8 million years, which is substantially more than the previous EWS studies mentioned), it is commonly hypothesized than silicate
weathering is the feedback that stabilize Earth climate (Walker, Hays and Kasting (1981) J Geophys Res 86, 9776; Berner and Caldeira (1997), Geol, 25, 955; François and Goddéris (1998), Chem Geol, 145, 177-212). The authors should explicitly say if they aim at tracking carbon-climate resilience due to silicate weathering feedback (or more generally feedbacks in geological carbon cycle), or only "shorter-term" climate resilience (for instance, the processes mentioned Page 3, lines 21–23), or both.

**You are right to say that it is important to be clearer about which long-term carbon cycle or climate processes may be implicated in our analysis, and the silicate weathering feedback is indeed the most important of these for the geological carbon cycle (along with long-term changes in the burial rates of organic carbon and ocean carbonate). As mentioned in page 11 line 6, due to the temporal resolution of our data we cannot resolve short-term processes (i.e. that take place over less then ~10 kyr), but in our revised manuscript we have**
**made this section clearer and be more explicit about which processes of the geological carbon cycle and long-**

**term climate system we expect to play a role (page 10 line 29) and include our interpretation that warming-driven changes in organic carbon burial are a likely pre-PETM mechanism in the Conclusions (page 17 line 14).**

It is of particular importance because of the timescale of considered perturbations of d13C and d18O: It is specified Page 4, lines 16-19 that the data are detrended in order to remove any long-term trends. This is indeed essential to get stationary time-series. But though it is explicitly said that bandwidth "is an important consideration", the only given information is that it is "adjusted heuristically for each dataset". More precision should be given on bandwidth value (is it constant along one given time-serie?) and above all because if it is less than the response time of carbon cycle (~100–200ky, François and Goddéris, 1998), then the indicators are not (or only partially) measuring the resilience due to its feedbacks. From the timescale of variations of smoothed records shown in Fig. 3, 4 S1 and S2 (red lines), I guess the bandwidth is actually in the order of 100ky. The choice of the timescale of the fluctuations to study and the feedbacks to investigate is up to the authors, but it should be specified, and conclusions can be drawn only for the focused feedbacks.

**You are also right that bandwidth choice is important in the context of process timescales as well, and that the original manuscript failed to mention the bandwidth choices for the main rolling window metrics – we found that the optimal Gaussian kernel bandwidth was 0.1 and have made this clear in the revised manuscript (page 12 line 8). This does not directly translate to a frequency limit, but as shown by the red line in Figures 2 and 3 this removes all of the secular trends and the long-term orbital-scale (~100+ kyr) variability. While this at first inspection would seem like we're filtering out what we're interested in (the long-term carbon cycle / climate system), with this methodology it is in fact the short-term noise that we are interested in as this short-term noise reveals the resilience of the longer-term processes. But as mentioned above, the data's temporal resolution places a lower limit on what timescale processes this method can reveal (in this case anything shorter than 10s of kyr). As a result, although we filter out the direct signal of longer-term processes, with this method this does not exclude these processes from affecting the results and interpretation. We have now added discussion of this to the background (page 10 line 29 onwards) and methodology (page 12 line 8).**

The section 3.2 (Binned Metrics) and Table 1 show that most of the indicators exhibit significant variations before/after the PETM and the ETM2. This suggest (as mentioned Page 7, lines 26–28) that the hyperthermal events are partly responsible for the loss of resilience of the carbon-climate system, or at least that these events are not simple perturbations followed by a relaxation towards the same "initial state": they induce some permanent or irreversible changes. Even if evidences of tipping points are lacking (as said Page 7, lines 27–28), I think it is an interesting results and should be highlighted in the conclusion, where there is no mention of this fact. I also wonder which component of carbon-climate system can be expected to undergo irreversible changes. If there is any "good candidate", it may be interesting to precise it.

**Our results do indeed clearly show increased and persistent long-term destabilisation following both hyperthermal events in our record, and this has implications for the early Eocene (as a destabilised carbon-climate system may have played a role in the subsequent repeated hyperthermals). In our original manuscript we mostly focused on the implications of destabilisation for the presence of tipping points preceding the two hyperthermal events in our record, but you are right to point out that the subsequent**

**persistence of the loss of resilience is important as well and we have made this clearer in the discussion (page 15 line 13 & line 30) and conclusions (page 16 line 28 & page 17 line 18) (as well as the processes likely to be implicated in this persistent loss of resilience, such as a dampened silicate weathering feedback in the Eocene relative to the Palaeocene).**

Minor specific remarks:

Page 4, line 15 and line 22: With the definition of autoregressive model as 'x(t+dt) = a*x(t) + e(t)', a constant timestep 'dt' is inherently necessary for the autoregressive coefficient 'a' is directly link to 'dt'. I wonder then how to fit an autoregressive model to the non-interpolated data? Just dividing 'a' by the local timestep is enough? It is not precised in the description of the 'generic_ews' function in R. Perhaps it is a "routine analysis" and is not worth to be precised, I can't really judge it.

**For the AR model time-step the R function simply assumes a constant time-step throughout the time-series (i.e. it takes the time-series length and divides by N to get the constant dt). Of course without interpolation the actual data points will slightly differ from this dt which will introduce some error (and is why the interpolated run is considered the default, despite the alternative problem discussed in the text that this introduces instead), but there is no systematic bias in the distribution real data time-steps which limits any systematic error as a result. The assumption of a constant time-step has been clarified (page 11 line 30 & page 12 line 14) and a plot of data time-steps provided in the Supplementary Material.**

Page 6, line 21: The word "divergence" (of standard deviation) may be misleading because the reader would firstly expect to find an "increase", which is in contradiction with the rest of the sentence (and the Figure). Perhaps it is preferable to substitute it for "decrease of standard deviation" or "reducing standard deviation". In addition, this decrease of SD is likely to be due to earlier "extreme" events (between 59.5 and 58.5Ma) that slip out of the rolling window when it reaches 57-56Ma. Indeed, with a 75% rolling window (Fig. S1), SD does not exhibit such a decrease. It does with a 25% rolling window probably because there are "extreme" events up to 57.5Ma. Therefore, this decrease of SD could be not linked to what happen immediately before the PETM, and not contradicting with the increase of AR1.

**We have clarified that d18O SD declines just before the PETM and discuss the likely reason why (page 14 line 19).**

Page 6, line 29: How to interpret a decrease in kurtosis? Shouldn't we except while approaching a tipping point more frequent extreme deviations, and then a higher kurtosis?

**We agree that one would expect kurtosis to increase prior to a tipping point as extreme data values become more common, but this is not considered to be as strong an indicator of resilience as other measures like variance or AR1. In this case it seems that the section in the d13C record between ~58-59.5 Ma has an especially high kurtosis (as shown in supplementary fig 1 especially by the 25% window) which then gradually falls as this period leaves the rolling window, which is likely masking any pre-PETM signal. Discussion of this feature has been added to the caption of Supp. Fig. 1.**

Page7, lines10–11: "but that the increase of AR(1) for both d18O and d13Carehighly significant (p=0)". Please add "for non-interpolated data" to be explicit.

**Clarified (page 15 line 11)**

Page 7, lines 13-15: There is another evidence that the values of d13C during the events (at least for the PETM) are partly responsible for the upward steps in the indicators: with a 25% rolling window (Fig. S2), both AR1 and SD show a downward step at 54Ma, exactly when the PETM leaves the rolling window (and after the ETM2 has come into the rolling window). However, it is true that the binned metrics clearly show steps that are not due to what happen during the events.

**The within-event data in the whole-dataset rolling window analysis does affect the results, but the binned data metrics that exclude the events confirm the shifts are present anyway – further detail on this contrast has been added (page 15 line 27).**

Page 8, lines 2-3: Same remark as for the decrease of SD before PETM: the decrease of jump intensity can be due to the "extreme" events between 59.5Ma and 57.5Ma.

**Unlike with the previous comment, the within-event data doesn't affect the DDJ model results as this method does not use a rolling window and all data are used and relevant.**

Supplementary Figures 1 and 2: How come than skewness is systematically positive? At least during PETM and ETM2, it seems there are more points beneath the red line than above. Moreover, in the binned metrics analysis (Table 1), skewness in systematically negative.

**There is indeed a mismatch between the d13C skewness as calculated on a rolling window (>~0 at all times and gradually increasing) and in the bins (<0 and decreasing for every bin). This is because the generic_ews function of the earlywarnings R package actually calculates and plots the absolute of the skewness – this is because skewness can increase or decrease when approaching a critical transition, and so the package plots the absolute in order to maintain consistency with the other indicators that should always increase. This means that the plotted skewness is in fact axis inverted, and matches the negative and declining values in Table 1. The events themselves skew positive (shown as a decrease in the plot) due to a detrending artefact that results in a temporary increase. We have updated the text, supplementary figures, and captions to make it clear the skewness plot is of the absolute values.**

**Additional Changes**

**In addition to the changes described above, we have also updated the p-values on the AR(1) rolling window and binned DFA-h in order to use arima model-generated surrogate records rather than bootstrapped datasets as a more appropriate null model choice (this led to only very minor changes in p-values though, with no impact on our results or discussion).**

[revised manuscript text omitted]